An automatic system for extracting figure-caption pair from medical documents: a six-fold approach

Chaki Jyotismita jyotismita@vit.ac.in
Department of Computational Intelligence, School of Computer Science and Engineering, Vellore Instiute of Technology , Vellore , India
Qu Rong
Electronic publication date: 2023 Jul 26
Publication date: 2023
Volume: 9
Electronic Location ID: e1452
Received 2023 Jan 13; Accepted 2023 Jun 1
Copyright: ©2023 Chaki
Copyright year: 2023
Copyright holder: Chaki
License: This is an open access article distributed under the terms of the Creative Commons Attribution License, which permits unrestricted use, distribution, reproduction and adaptation in any medium and for any purpose provided that it is properly attributed. For attribution, the original author(s), title, publication source (PeerJ Computer Science) and either DOI or URL of the article must be cited.
License URL: https://creativecommons.org/licenses/by/4.0/

Keywords: A-torus wavelet transform, Figure-caption pair, Maximally Stable Extremal Regions connected component, Bounding box, MLP

Funding: The authors received no funding for this work.

==============================
Background

Figures and captions in medical documentation contain important information. As a result, researchers are becoming more interested in obtaining published medical figures from medical papers and utilizing the captions as a knowledge source.

Methods

This work introduces a unique and successful six-fold methodology for extracting figure-caption pairs. The A-torus wavelet transform is used to retrieve the first edge from the scanned page. Then, using the maximally stable extremal regions connected component feature, text and graphical contents are isolated from the edge document, and multi-layer perceptron is used to successfully detect and retrieve figures and captions from medical records. The figure-caption pair is then extracted using the bounding box approach. The files that contain the figures and captions are saved separately and supplied to the end useras theoutput of any investigation. The proposed approach is evaluated using a self-created database based on the pages collected from five open access books: Sergey Makarov, Gregory Noetscher and Aapo Nummenmaa’s book “Brain and Human Body Modelling 2021”, “Healthcare and Disease Burden in Africa” by Ilha Niohuru, “All-Optical Methods to Study Neuronal Function” by Eirini Papagiakoumou, “RNA, the Epicenter of Genetic Information” by John Mattick and Paulo Amaral and “Illustrated Manual of Pediatric Dermatology” by Susan Bayliss Mallory, Alanna Bree and Peggy Chern.

Results

Experiments and findings comparing the new method to earlier systems reveal a significant increase in efficiency, demonstrating the suggested technique’s robustness and efficiency.

Introduction

The utilization of digital health recording systems is slowly increasing, particularly amongst major hospitals, and in the coming years, the e-health platform will be fully available. A significant amount of paper-based health records is still stored in hospital health record repositories. These are the long history of each patient’s medical assessment and, therefore, these are reliable sources of medical research and enhanced patient care (Beck et al., 2021). Due to the significance of these paper records, few hospitals have begun to digitize them as image files where the patient ID is the only reliable key to accessing them, but most hospitals have preserved them as they are. This is because of the high digitization costs and also the comparatively low benefit of records that can only be accessed by patient identification. However, the real purpose of computerizing paper documents is to provide them with functionality so that they can be utilized in medical research such as extracting similar cases between them. If it is difficult to identify realistic possible solutions to this problem, large amounts of these paper-based clinical records will soon be lost in book vaults and the coming years may be destroyed. Therefore, a challenge is faced of designing a better system that is easy to run and can easily integrate the paper-based large history of clinical records into the e-health climate.

For clinical or medical records, medical images and captions deliver essential information. Medical data includes medical papers, dissertations, and theses, as well as information on a variety of clinical trials and life sciences studies. Typically, this information is available as scanned data. For all valuable information, these scanned medical images cannot be read manually. Manually, analyzing and extracting information from all the scanned files available in the medical data repository will take years. This is among the main reasons why large data analysis and visualization with the utilization of machine learning are needed. The data is unavailable as a lot of information still in the form of scanned data and these formats cannot be efficiently interpreted. Using the internet, researchers may access medical data in a variety of formats. Some of these involve single column versus double-column layouts, layout variance, etc. Extracting all the necessary information from medical scanned images becomes almost difficult. Although machine learning can manage vast amounts of data in conjunction with other AI approaches, the complexities of extracting data from scanned images go back generations (Wu et al., 2021). Medical image extraction, distinguishing the pixels of organs or diseases from the context of the medical record, is one of the most difficult challenges of medical image processing that is to provide crucial information on the structures and sizes of these organs. Thus, an important new technique has been implemented in this study to extract medical figures and captions from medical reports. Because of the complex and varied nature of medical publications and the differences in figure form, texture, and material, such extraction is not a simple process. Because medical figures frequently contain several image panels, research has been done to classify compound figures and their constituent panels.

Several researchers have proposed different systems (Li, Jiang & Shatkay, 2018; Li et al., 2017) to extract graphics from medical scanned images automatically by applying different technologies available. Previous systems have been based on traditional techniques like filters for edge detection (Singh et al., 2012; Li, Jiang & Shatkay, 2019; Somkantha, Theera-Umpon & Auephanwiriyakul, 2010; Naiman, Williams & Goodman, 2022; Trabucco et al., 2021) and mathematical techniques (Senthilkumaran & Vaithegi, 2016; Piórkowski, 2016; Rajinikanth et al., 2018; Li et al., 2021). Then, for a long time, machine learning techniques are used to extract hand-crafted attributes have become a dominant methodology to extract medical figures from the medical document scanned (Jiang et al., 2020; Xia et al., 2017; Espanha et al., 2018). The design and implementation of these features have always been the main concern for the development of such a program, and the complexity of these methods has been considered as a major limitation for their deployment. Later, deep learning techniques came into the picture as a result of hardware improvements and began to show their significant abilities in image processing tasks. The impressive skill of deep learning methods has made them a preferred choice for extracting medical graphics from medical scanned documents (Zhou, Greenspan & Shen, 2017; Pekala et al., 2019; Fritscher et al., 2016; Dalmış et al., 2017; Moeskops et al., 2016). The extraction of medical figures created on deep learning methods has established considerable attention, particularly in the previous few years, and it highlights the need for a comprehensive review of it. Various platforms, like the Yale Image Finder, BioText, Open-I, askHermes, and the GXD database purpose to allow users to find suitable medical figures and captions from medical documents (Demner-Fushman et al., 2012; Sanyal, Chattopadhyay & Chatterjee, 2019; Xu, McCusker & Krauthammer, 2008; Yu, Liu & Ramesh, 2010). Some researchers have proposed some methods to extract medical figures along with their caption from medical documents (Li, Jiang & Shatkay, 2018; Demner-Fushman, Antani & Thoma, 2007; Pavlopoulos, Kougia & Androutsopoulos, 2019; Lopez et al., 2011). Though, the primary step toward this aim, viz., extracting figure-associated caption pairs from medical documents is neither well-studied nor so far well-addressed. Thus, an efficient and new technique is introduced to extract figures and captions from medical publications.

The contributions of this study are as follows:

• In this article, an effective and new six-fold approach is presented to extract figures and related captions from scanned medical documents. In contrast to previous methods, the raw graphical objects stored in the scanned files are not examined directly by the proposed method.

• The edges are extracted from the scanned document using a-torus wavelet transform. Then, it distinguishes the text element from the scanned file’s graphical element and applies maximally stable extremal regions connected component analysis to the text and graphical element to identify individual figures and captions. Text and graphics are separated by using multi-layer perceptron. It is trained to recognize the text parts from the scanned file so that the graphical content parts are identified easily. The bounded box concept is used to create separate individual blocks for every figure-caption pair.

• The proposed system is tested using a self-created dataset comprised of the pages from the five open access books titled “Brain and Human Body Modelling 2021” by Sergey Makarov, Gregory Noetscher and Aapo Nummenmaa, (Makarov, Noetscher & Nummenmaa, 2023) “Healthcare and Disease Burden in Africa” by Ilha Niohuru (Niohuru, 2023), “All-Optical Methods to Study Neuronal Function” by Eirini Papagiakoumou (Papagiakoumou, 2023), “RNA, the Epicenter of Genetic Information” by John Mattick and Paulo Amaral (Mattick & Amaral, 2023) and “Illustrated Manual of Pediatric Dermatology” by Susan Bayliss Mallory, Alanna Bree and Peggy Chern (Mallory, Bree & Chern, 2005).

The rest of the article presents the details of the method and validates its efficiency through a sequence of experimentations. ‘Materials & Methods’ deliberates the related works; in ‘Results’, the proposed methodology is presented; ‘Discussion’ presents the experimentations utilized to measure the performance of the proposed technique, together with the results attained by the approach and by other previously established methods utilized for comparison; ‘Conclusions’ includes the discussion; while Section 6 concludes and summaries directions for future work.

Materials & Methods

This study aims to extract figures and captions from medical documents. Figure 1 summarizes the full framework for the proposed six-fold method. Throughout this research, the classification of text and graphics is done by classification at the level of the connected component (CC) within a single framework. The processing flow includes two main stages after binarization, edge extraction, and CC extraction. Those are text block classification and graphics block classification. The methodology contains six basic steps: Pre-processing, Edge extraction using a-torus wavelet transform, CC extraction using Maximally Stable Extremal Regions (MSER), identification of graphics blocks, detection of caption, and identifying figure-caption pair.

Figure 1 Framework of the proposed approach.

Pre-processing

The pages collected from medical documents (in PDF format) are first converted to images. These images may be in grayscale or color format. In this step, if the input scanned image is a color image, a weighted sum of red, green, and blue components is used to convert it to grayscale. Eq. (1) depicts the conversion process. (1) gx,y=0.2989×R+0.5878×G+0.1140×B.

Where R, G, and B are red, green, and blue color channels of a color image respectively.

The next step of pre-processing is to enhance the grayscale image by reducing the noise. Different grayscale image enhancement methods are published in the literature. In the proposed method, a smoothing filter is applied to improve the transformed grayscale document image g(x, y) followed by unsharp filtering.

The grayscale image is transmitted through an appropriate filter to boost the image quality. The peak signal to noise ratio (PSNR) is utilized as a parameter to determine the appropriateness of the filter. PSNR is a measure of image quality. The higher the output of the PSNR, is better the quality of the image. The filter that offers the highest PSNR is therefore selected in this step. In this study, three filters are utilized for smoothing namely: wiener filter, median filter, and low pass filter.

Wiener filter: This filter is intended to preserve the image in such a way that there should be less square error amongst the original and restored image.

Median filter: This filter preserves each pixel of its neighborhood pixels with a median. This significantly reduces the pepper and salt noise.

Low pass filter: This filter preserves each image pixel with its neighborhood pixel mean.

The mask displayed in Fig. 2 is used to get an unsharp filtered image from the smoothened image s(x, y). This leads to the image u(x, y) which can be used to extract enhanced edge information.

Figure 2 Unsharp filter mask.

The PSNR for an image can be calculated by utilizing Eq. (2). (2) PSNR=10× log10G2MSE.

Where “G” is an image’s overall gray level value and MSE is a mean square error amongst the original and the enriched image. Let g(x, y) and u(x, y) are of size P × Q, The MSE can be calculated by using Eq. (3). (3) MSE=1P×Q∑x=1P ∑y=1Qgx,y−ux,y2.

Edge detection with the undecimated wavelet transform

After enhancing the scanned image, the next step is to detect edges from u(x, y) using a-torus undecimated wavelet transform.

A-torus undecimated wavelet transform

The fast-multi-scale edge detector of Mallat [33] is implemented utilizing an undecimated form of the wavelet transformation called the a-torus algorithm. The discrete method to the wavelet transform may be accomplished using a modified version of the a-trous algorithm. The technique may deconstruct an image (or a signal) into an approximation signal (A) and a detail signal (D) at a scale; the detail signal is referred to as a wavelet plane, and it has the same dimension as the original image. With every decomposition step in the a-trous algorithm, the low-pass and high-pass wavelet filters are spread, and the convolution is done with no sub-sampling. N + 1 (N detail images plus a single image approximation) images of the same size are generated by a-trous algorithm, where N is the number of levels of decomposition. For several different reasons, this algorithm is very efficient: (1) it is translation-invariant (a shift in the input simply moves the coefficients), (2) the discrete transform values are precisely determined at each pixel position without any interpolation, and (3) the correlation between scales is easily manipulated because of its inherent structure. In this study, the detail coefficients are determined as differences amongst consecutive approximation coefficients. With this concept, reconstruction is simply the sum of all the detailed matrices of the coefficient and the final matrix of the approximation coefficient. For an N-level decomposition, the reconstruction formula is then expressed by using Eq. (4). (4) Reconstructed_Image=AN+ ∑a=1NDN.

Edge detection

There are several options of wavelet basis functions, and it is very critical to choose a perfect basis that can be used for edge detection. The mother wavelet, in particular, should be symmetrical. Symmetry is significant since in this study a smooth image is differentiated, and thus a lack of symmetry means that edge position will alter as the image is successively smoothed and differentiated. With the correct wavelet selection, at a specified scale, the edge positions correspond to the maximum modulus of the wavelet transform.

The cubic wavelet B-spline (B3) perfectly fits with the proposal as it is symmetrical and easy to derive. A degree N-1 spline is the convolution of N box functions in particular. This study (1/2, 1/2) is transformed four times to produce the low-pass filter coefficients of the B3 as shown in Eq. (5). (5) B3=12,12∗12,12∗12,12∗12,12=1161,4,6,4,1.

The above-mentioned Eq. (5) is extended to two dimensions to produce a low-pass filter kernel (KLPF) suitable for the utilization with the a-torus algorithm as shown in Eq. (6). (6) KLPF=B3TB3=12561643128164125616411633211616431283329643323128164116332116164125616431281641256.

This kernel is interesting because left bit-shifts can be substituted for the costly division operations as the denominators are all powers of two. There is the ability with this edge detection algorithm to monitor the behavior of the edge detection scheme by integrating scale into the overall system. There is no concept of scale with traditional edge detectors, but with a wavelet model, by regulating the number of wavelet decompositions, the scale can be adjusted as desired. Scale determines the significance of the edges that are observed. The various balance between image resolution and edge scale may result from various edge detected images such as high resolution and small scale (minimal number of wavelet decompositions) outcomes in relatively noisier and more discontinuous edges. Low resolution combined with large scale, therefore, can lead to undetected edges. Sharper edges are more prone to be maintained by transforming the wavelet into subsequent scales, while lower edges are attenuated as the scale rises.

Connected Component (CC) extraction using MSER

A novel CC-based region detection algorithm that uses MSER is used in this article. Several co-variant regions, named MSERs, are extracted from an image by the MSER algorithm: an MSER is a stable connected image component. MSER is built on the concept of taking areas that remain substantially constant across a wide variety of threshold. All pixels below a certain threshold are white, whereas all pixels over or equal to that threshold are black. If there were a succession of thresholded images It with frame t matching to threshold t, a black image can be observed at first, then white dots corresponding to local intensity minima form and get larger. These white dots will ultimately merge, resulting in a white image. The set of all extremal regions is the set of all connected components in the sequence. The term extremal refers to the fact that all pixels within the MSER are either brighter (bright extremal regions) or darker (dark extremal regions) than all pixels on its outer boundary.

This study aims to find out the graphical element along with the associated caption from the medical document. The bounding box region property is used to filter the connected components of text and graphics elements. The bounding boxes are expanded to the left, right, and up, downside so that some overlapped boxes are formed. Then the overlapped boxes are merged to form a new bounding box. Thus, reducing the number of bounding boxes. Also, a threshold is set to eliminate the small regions as the area of the medical graphics elements is not very tiny. The area of the regions less than the threshold is ignored for further processing.

Graphic objects detection

The contents of each bounding box are extracted and the percentage of text concerning the bounding box size is measured. For this purpose, some morphological operations like dilation and filling are used to enhance the characters of each bounding box so that it can be recognized by the text classifier. After that, some important features are computed from each enhanced character, and a multi-layer perceptron (MLP) is utilized to recognize the text. Extracted features are numerical values and are stored in arrays. MLP can be trained with these numerical values. The most significant features for character recognition used in this proposed approach are as follows.

1. H-position: Horizontal position which is the count of pixels from the left edge of the character to the centroid of the minutest bounding box with all character pixels inside.

2. V-position: Vertical position which is the count of pixels from the bottom of the character bounding box to the above box.

3. Width: The width of the bounding box, in pixels.

4. Height: The height of the bounding box, in pixels.

5. Total pixel: The total number of character image pixels.

6. Mean H-position: Mean horizontal position of all character pixels relative to the centroid of the bounding box and divided by the width of the box. This feature has a negative value if the image is lefty-heavy, for eg., letter F.

7. Mean V-position: Mean position in the vertical direction of the entire character pixels corresponding to the centroid of the bounding box and divided by the height of the box.

8. Mean SQ-H: The horizontal pixel’s mean squared value of the distances as computed in VI above. This feature will have a larger value for images whose pixels are broadly divided in the horizontal direction, for eg., letters M or W.

9. Mean SQ-V: The vertical pixel’s mean squared value as calculated in VII above.

10. Mean PROD-HV: The mean product of the horizontal and vertical distances for every character pixel as calculated in 6 and 7 above. This feature has a negative value for the top left to bottom right diagonal lines and a positive value for the bottom left to top right diagonal lines.

11. H-variance: For each character pixel, the mean value of the squared horizontal distance adjusts the vertical distance. This calculates the correlation of the horizontal variance with the vertical position.

12. V-variance: For each character pixel, the mean value of the squared vertical distance adjusts the horizontal distance. This calculates the correlation of the vertical variance with the horizontal position.

13. Mean V-edge: The mean number of edges (a character pixel to the immediate right of either the character boundary or a non-character pixel) come across while doing scans from left to right at all vertical positions within the bounding box. This feature differentiates amongst letters like “L” or “I” and letters like “M” or “W”.

14. The sum of the edge positions in the vertical direction is described in XIII above. If there are more edges at the top of the bounding box, as in the letter “Y”, this feature would give a higher value.

15. Mean H-edge: The mean number of edges (a character pixel to the immediate above of either the image boundary or a non-character pixel) come across while doing scans over the entire horizontal positions within the box of the character from the bottom to top.

16. The sum of edges horizontal positions is described in 15 above.

With the purpose of standardization of the size of sub-images, the sub-images must be cut close to the border of the character before extracting the features from the characters. The standardization of the image is achieved by determining the maximum column and row with 1s and the highest point increasing and decreasing the counter till the black space is reached, or the line with all 0s. This method is demonstrated in Fig. 3 where a character “P” is being cropped and resized.

Figure 3 (A) Determining X-min, Y-min, x-max, and Y-max to crop the character, (B) Cropped and resized image.

The aforementioned 16 features (input of MLP) are utilized to identify the character from the bounding box.

Then the percentage of bounding box area occupied by the text is computed by dividing the total area of the bounding box by the total area of the character set or text inside the bounding box. A threshold is set to classify between the text and graphics blocks. If the percentage of text is less than the threshold, the bounding box object is considered as the graphics element of the document page.

Detection of caption

Captions that can assist to locate the related figures are first identified utilizing the header prefixes of the caption like Fig, FIG, Figure, or FIGURE. For every possible figure caption, its position is recorded as it occurs in the document image file, and primarily its bounding box width is set to be the length of the first text line of the caption. A figure is generally located either immediately above its caption’s topmost line, below its bottom line, or on its caption’s side—where the figure’s bottom or top are aligned with the bottom or the caption’s top, correspondingly. The figure area naturally lies near the caption area. Therefore, the figure position is used to recognize possible caption positions.

The continuous text block succeeding the possible header is identified by utilizing the character recognition algorithm discussed in ‘Graphic Objects Detection’. The caption text block is demarcated as an arrangement of characters, where the last character ends with a period and where the last character is trailed by a vertical gap or break (i.e., by a gap whose height beats the regular gap amongst body text lines).

Figure-caption pair extraction

After identifying the caption and figure separately, the next step is to combine the bounding boxes of the figure and associated captions so that they can be extracted as a single element. For that first four parameters are extracted from the bounding boxes of the figure and caption: x-position, y-position, width, and height of the bounding box. Three individual situations are noticed as mentioned in Table 1.

Table 1 Extracted parameters from bounding box.

	

After merging the bound boxes of the figure and associated caption, the full figure and its related caption are then preserved as part of the output files together with their bounding boxes on the associated document page which indicates their respective location.

Results

The output attainable by the proposed method in recognizing the graphic elements along with the caption from the medical document will be provided in this section. Using MatLab R2017a, the proposed technique was implemented on Intel Core i7 CPU 3.40 GHz. The effectiveness of the proposed approach is tested on a self-created dataset one comprised of 1563 pages in image format collected from five free downloadable (open access) medical books: “Brain and Human Body Modelling 2021” by Sergey Makarov, Gregory Noetscher and Aapo Nummenmaa, “Healthcare and Disease Burden in Africa” by Ilha Niohuru, “All-Optical Methods to Study Neuronal Function” by Eirini Papagiakoumou, “RNA, the Epicenter of Genetic Information” by John Mattick and Paulo Amaral and “Illustrated Manual of Pediatric Dermatology” by Susan Bayliss Mallory, Alanna Bree and Peggy Chern. These books are selected as these books are rich source of figure-caption pair.

Table 2 provides the statistics of the dataset. There is a total of 912 figure-caption pairs observed in the dataset.

Table 2 Statistics for each dataset used in this study.

Book name	No. of pages	No. of figures	No. of captions	No. of figure-caption pair	
Brain and human body modelling, 2021	165	78	78	78	
Healthcare and disease burden in Africa	142	171	171	171	
All-optical methods to study neuronal function	424	127	127	127	
RNA, the Epicenter of Genetic Information	400	100	100	100	
Illustrated Manual of Pediatric Dermatology	432	436	436	436	

Pre-processing

In this step, first, the freely accessible medical books are downloaded in PDF format and each page of those books are converted to images. After that images are converted into grayscale (Fig. 4).

Figure 4 (A) Original scanned image, (B) Converted grayscale image.

Image source credit: illustrated Manual of Pediatric Dermatology by Susan Bayliss Mallory, Alanna Bree and Peggy Chern, CC BY-NC-ND 4.0 (https://creativecommons.org/licenses/by-nc-nd/4.0/).

The output of the smooth grayscale image using three filters as mentioned in ‘Pre-processing’ is shown in Fig. 5. After several experimentations using the first dataset, the size of the filter mask is set to 7 × 7 pixels.

Figure 5 Output of the smoothened image using three filters: (A) Wiener filter, (B) Median filter, (C) Low pass filter.

Image source credit: Illustrated Manual of Pediatric Dermatology by Susan Bayliss Mallory, Alanna Bree and Peggy Chern, CC BY-NC-ND 4.0 (https://creativecommons.org/licenses/by-nc-nd/4.0/).

Figure 6 shows the enhanced image outputs after applying the unsharp mask as described in ‘Pre-processing’ to Figs. 5A, 5B and 5C.

Figure 6 Enhanced (sharpen) version of Fig. 5(A), (B), and (C).

Image source credit: Illustrated Manual of Pediatric Dermatology by Susan Bayliss Mallory, Alanna Bree and Peggy Chern, CC BY-NC-ND 4.0 (https://creativecommons.org/licenses/by-nc-nd/4.0/).

For each test image, the PSNR is calculated between the grayscale and three types of enhanced sharpen images. The rounded average PSNR values using those three types of filters are as follows: Wiener = 65, Median = 32, and Low pass = 41. Hence, the Wiener filter is used in this study to enhance the grayscale image.

Edge detection

After enhancing the image, the next step is to detect the edges using a-torus undecimated wavelet transform as described in ‘Edge Detection with the Undecimated Wavelet Transform’. The output of the detected edges of sample Fig. 6A is shown in Fig. 7.

Figure 7 The output of edge detection.

Text and graphics bounding box detection using MSER connected component

The bounding box is detected for both text and graphics elements from the edge image using MSER connected component feature as described in ‘Connected Component (CC) Extraction using MSER’. After several experimentations using the first dataset, the expansion amount is set to 5 pixels to merge the small bounding boxes into one and after merging, the threshold to ignore small bounding box content is set to 7,000 pixels. The output of the finally detected bounding boxes (yellow boxes) from Fig. 7 is shown in Fig. 8. The detected boxes are projected onto the original scanned image for better understanding.

Figure 8 Detected bounding boxes.

Image source credit: Illustrated Manual of Pediatric Dermatology by Susan Bayliss Mallory, Alanna Bree and Peggy Chern, CC BY-NC-ND 4.0 (https://creativecommons.org/licenses/by-nc-nd/4.0/).

Detection of graphic element

From the detected bounding boxes, only the graphics elements are extracted by using the MLP approach as mentioned in ‘Detection of Graphic Element’. After several experimentations using the first dataset, the threshold used to distinguish between text and graphics bonding box content is set to 20%. Figure 9 depicts the output of the detected graphics blocks.

Figure 9 Detected graphic bounding box.

Image source credit: Illustrated Manual of Pediatric Dermatology by Susan Bayliss Mallory, Alanna Bree and Peggy Chern, CC BY-NC-ND 4.0 (https://creativecommons.org/licenses/by-nc-nd/4.0/).

Figure-caption pair extraction

The caption of the associated figure is detected (Fig. 10A) by using the location of the figure as well as the caption in the scanned page as described in ‘Detection of Caption’ and ‘Figure-caption Pair Extraction’. After detecting the associated caption of the graphic element, both bounding boxes are finally merged to be detected as a single element as shown in Fig. 10B.

Figure 10 (Left) Detected caption and graphic bounding box, (Right) merged bounding box of detected figure and the associated caption.

Image source credit: Illustrated Manual of Pediatric Dermatology by Susan Bayliss Mallory, Alanna Bree and Peggy Chern, CC BY-NC-ND 4.0 (https://creativecommons.org/licenses/by-nc-nd/4.0/).

The results obtained by using the created dataset and the proposed method are presented in Table 3 (precision, recall, and F-score) and are compared with the methods used by previous researchers, when performing (A) figure extraction, (B) caption extraction, and (C) figure-caption pair extraction.

Table 3 Analysis of the proposed method with previous approaches.

	Method	Precision (%)	Recall (%)	F-score (%)	
Figure Extraction	Approach used in Li, Jiang & Shatkay (2018)	90.24	88.10	85.64	
Approach used in Clark & Divvala (2015)	71.09	68.21	70.74	
The approach used in Choudhury et al. (2013)	88.12	61.45	74.62	
The approach used in Naiman, Williams & Goodman (2022)	91.08	90.24	88.84	
Proposed approach	96.73	94.21	95.86	
Caption Extraction	Approach used in Li, Jiang & Shatkay (2018)	73.16	84.74	79.59	
Approach used in Clark & Divvala (2015)	33.10	40.34	37.27	
The approach used in Choudhury et al. (2013)	81.52	76.95	75.62	
The approach used in Naiman, Williams & Goodman (2022)	90.14	89.76	85.29	
Proposed approach	92.87	87.14	89.94	
Figure-Caption Pair Extraction	Approach used in Li, Jiang & Shatkay (2018)	88.59	83.17	86.13	
Approach used in Clark & Divvala (2015)	51.14	60.98	54.11	
The approach used in Choudhury et al. (2013)	64.17	61.30	62.84	
The approach used in Naiman, Williams & Goodman (2022)	89.76	85.49	87.18	
Proposed approach	91.76	88.12	90.17	

Precision is a measure of how many correct positive predictions are produced (true positives). Precision is an excellent emphasis if the goal is to reduce false positives. In this study, as minimal mistake in the figure-caption pair as feasible is expected. But I don’t want to overlook any crucial figure-caption pair. In such instances, it is reasonable to expect it to strive for maximum precision. With respect to the figure-caption pair, it can be represented by using Eq. (7). (7) Precision=True PositiveTrue Positive+False Positive=No. of correctly predicted positive instancesNo. of total positive predictions made=No. of correctly predicted figure-caption pairTotal no. of predicted figure-caption pair.

For only the figure extraction, only the caption extraction and the figure-caption pair extraction from the dataset, the precision values are 96.73%, 92.87% and 91.76% respectively.

Recall is a measure of how many positive cases the classifier predicted correctly out of all the positive cases in the data. Recall is critical in fields like healthcare, where we wish to limit the possibility of missing positive instances (predicting false negatives). In many circumstances, missing a positive case has a considerably higher cost than incorrectly classifying something as positive. With respect to the figure-caption pair, it can be represented by using Eq. (8). (8) Recall=True PositiveTrue Positive+False Negative=No. of correctly predicted positive instancesTotal no. of positive instances in the dataset=No. of correctly predicted figure-caption pairTotal no. of figure-caption pair in the dataset.

For only the figure extraction, only the caption extraction and the figure-caption pair extraction from the dataset, the recall values are 94.21%, 87.14% and 88.12% respectively.

F1-Score is a metric that combines precision and recall. It is commonly referred to as the harmonic mean of the two. Harmonic mean is just another approach to calculate an “average” of numbers, and it is typically touted as being more suited for ratios (such as precision and recall) than the standard arithmetic mean. It can be represented by using Eq. (9). (9) F-score=2×Precision×RecallPrecision+Recall.

For only the figure extraction, only the caption extraction and the figure-caption pair extraction from the dataset, the F-score values are 95.86%, 89.94% and 90.17% respectively.

These results demonstrate that the proposed method offers an accurate and reliable means of extracting figures from image documents; it is especially suited in the field of digitization of medical documents, where related articles can cover a wide range of years of publication. In terms of the extraction of the caption, the efficiency of the proposed technique is decreasing compared to the extraction of the figure. The reason behind this is the conversion of the pages from PDF to image, which often leads to inconsistencies that make it harder to identify captions. Thus, for caption extraction, the approach used in Li, Jiang & Shatkay (2018) shows a recall level of 84.74%, while the approach used in Clark & Divvala (2015) produces the recall is at 40.34% and the approach used in Choudhury et al. (2013) shows a recall level of 76.95%. In contrast, the proposed method displays a meaningfully higher recall after facing these hurdles, namely, 88.55% on the same dataset. The proposed system is, therefore, more robust and effective in recognizing captions in a wide range of scanned files compared to existing schemes. Finally, the results are analyzed for the combined task of extraction of the figure-caption pair. Specifically, in this task, all three approaches show lower efficiency as it involves proper extraction of both the figure and its associated caption. Once again, these results confirm and support the proposed system as an efficient and reliable method to extract pairs of figure-caption.

Figures 11A–11C demonstrates some examples of figures and captions extracted by the proposed method. The proposed system properly extracts the figure and its related caption both in the simpler situations where there is only one figure with caption present in a page (Fig. 11C), and in more difficult situations, where there are more than one figure with many boundaries and texts placed vertically as a single figure with one caption (Fig. 11a) and different figures placed horizontally as a single figure (Fig. 11B) with a single caption. On the other hand, Figs. 11A1–11C1, 11A2–11C2, and 11A3–11C3 show the (inappropriate) extraction achieved over the same scanned pages by the other three approaches (Li, Jiang & Shatkay, 2018; Clark & Divvala, 2015; Choudhury et al., 2013). Those approaches directly handle the figure objects the scanned file to extract figures, and as such misinterpret some of the figures encoded in a complex document structure—even when the document structure is simple (e.g., Figs. 11C2 and 11C3).

Figure 11 Test Output Analysis.

(A, A1, A2, A3) Image source credit: All-Optical Methods to Study Neuronal Function by Eirini Papagiakoumou, CC BY-NC-ND 4.0, (B, B1, B2, B3) Image source credit: Illustrated Manual of Pediatric Dermatology by Susan Bayliss Mallory, Alanna Bree and Peggy Chern, CC BY-NC-ND 4.0, (C, C1, C2, C3) Image source credit: RNA, the Epicenter of Genetic Information by John Mattick and Paulo Amaral, CC BY-NC-ND 4.0.

It is to be noted that while the proposed approach extracts most of the figures and captions properly, but there are still some cases where the extraction is not correct. The figure-extraction task is mainly challenging where small graphical objects are present in the figure and thus making it is tough to identify whether the boundaries of the figure include or exclude the next figure contents. Figure 12 shows such two situations, where the proposed technique misidentifies the figure region. A better and satisfactory metric should take into consideration for the definite retrieval of contents instead of the coordinates of the bounding box; we plan to create such a metric as part of the future work.

Figure 12 Example of inexact extraction by the proposed method.

(Left) Image source credit: All-Optical Methods to Study Neuronal Function by Eirini Papagiakoumou, CC BY-NC-ND 4.0, (right) Image source credit: RNA, the Epicenter of Genetic Information by John Mattick and Paulo Amaral, CC BY-NC-ND 4.0.

Discussion

This study is mainly done to extract the figure-caption pair from a document. In this study only the biomedical documents are used, but this approach can be applied to other documents as well. In this methodology, six-folds are proposed. First is pre-processing where the PDF pages are converted to the image and then the image is converted to grayscale. After that the image is enhanced by reducing the noise. To reduce the noise, wiener filter is used. The next step is to detect the edge by using a-torus undecimted wavelet transform. In the third step, a novel connected component-based region detection algorithm is used that uses MSER. Several co-variant regions, named MSERs, are extracted from an image by the MSER algorithm: an MSER is a stable connected image component. The next step is for graphic object detection. The contents of each bounding box (created from the connected component) are extracted and the percentage of text concerning the bounding box size is measured. For this purpose, a multi-layer perceptron is used. A threshold is set to classify between the text and graphics blocks. If the percentage of text is less than the threshold, the bounding box object is considered as the graphics element of the document page. The fifth step is to detect the caption. The figure area naturally lies near the caption area. Therefore, the figure position is used to recognize possible caption positions. The next and final step is to extract the figure-caption pair. For this step the detected bounding box of figure and caption is merged and then preserved as the output file. This method is compared with other methods in the literature to prove the efficacy of the method. A self-created datasets are used here for the experimentation purpose where the pages are collected from five freely accessible (open access) medical documents. For only the figure extraction from the dataset, the precision, recall and F-score values are 96.73%, 94.21% and 95.86% respectively. For only the caption extraction from the dataset, the precision, recall and F-score values are 92.87%, 87.14% and 89.94%, respectively. For the figure-caption pair extraction from the dataset, the precision, recall and F-score values are 91.76%, 88.12% and 90.17% respectively.

Conclusions

In this study, a new and efficient six-fold method is presented for extracting figures, captions, and figure-caption pairs from freely accessible medical documents. Previous techniques generally perceive figures by directly finding the contents present in the scanned file and handles the graphical objects encoded in it. When the figure and the document structures are complex, this approach sometimes leads to inappropriate extraction and that is a common phenomenon within medical publications. On the other hand, the proposed approach first fully separates the text contents from the graphical contents of the image file using multi-layer perceptron and aims to extract figure-caption pair using the bounding box concept. It applies maximally stable extremal regions connected component analysis to the image to detect figures, and separately finds the text portion for captions that lie in the neighborhood of the identified figures.

To test the system and compare it to state-of-the-art approaches, a self-created dataset is used. Pages are converted to images from five open access medical books and perceived as the testing dataset. Extensive experiments and results validate that the proposed system is highly efficient concerning the precision, recall, and F-score. Furthermore, the proposed system holds its good performance over documents that varies broadly in style, topic, overall organization, and publication year. Thus, it is ready to be useful in practice.

As part of future work, a new assessment metric will be planned that will make up for both figures and captions’ actual retrieved contents, instead of merely correctly identifying bounding box locations. Some deep learning-based methods can be integrated in future to segment the figure and caption from the document.

Additional Information and Declarations

Competing Interests

Author Contributions

Data Availability

Jyotismita Chaki is a Section Editor of PeerJ Computer Science.

Jyotismita Chaki conceived and designed the experiments, performed the experiments, analyzed the data, performed the computation work, prepared figures and/or tables, authored or reviewed drafts of the article, and approved the final draft.

The following information was supplied regarding data availability:

The figure caption recognition is available at Github and Zenodo: https://github.com/Jyotismita-1/figure-caption.

Jyotismita chaki. (2023). figure-caption recognition. https://doi.org/10.5281/zenodo.7527836.

The data is available at figshare: Chaki, Jyotismita (2023). Medical dataset. figshare. Dataset. https://doi.org/10.6084/m9.figshare.21894681.v1.

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
