# Peer review of "An automatic system for extracting figure-caption pair from medical documents: a six-fold approach"

_PeerJ Computer Science, doi:10.7717/peerj-cs.1452_

## Round 0.1 · original submission · Major Revisions

The authors are encouraged to fully justify the contributions with more experimental analysis and technical details of the datasets as the reviewers recommended.

Reviewer 1 ·

Basic reporting

The author propose an interesting method for figure and caption extraction from medical documents. To do that, the author first apply edge detection on the scanned documents. Next, connected component are used to identify the connected regions within the document. Specific features are used to identify the graphic content and a heuristic method is used for identify caption. The method description is clear. However, the experiment only tested on two scanned books rather than test on a large number of biomedical documents. The author needs to provide more information about their experiments before the publication.

Experimental design

The method only tested on two scanned books. As the book typically follow the same layout for presenting figure and caption pairs, the author should test their method on a large number of biomedical publications.
Moreover, the exact number of figures, captions, and figure-caption pairs should be reported for those two books.
The evaluation metrics, precision, recall and f-score should also be further explained.

Validity of the findings

The author need either provide more details about current experiments on two books or provide more experiments on a large number of biomedical publications to prove the impact of the work.

Additional comments

Minor comments:
1. The section in the paper is not numbered. So, please number them first before referring to them.
2. The edge detector of Mallat needs a reference.
3. The MSER algorithm mentioned by the author cannot find in the draft.

Reviewer 2 ·

Basic reporting

In this paper, the authors presented a six-fold approach for extracting figures and captions from medical documents and pairing the figure extracted with the corresponding captions. The six steps of the proposed method include preprocessing, edge detection, connected component extraction, graphic objects detection, caption detection and figure-caption pair extraction.

I have the following comments for authors to consider.
The writing needs to be enhanced. Some examples where the language could be improved include lines 15, 24, 25, 56, 346. I suggest the authors have a colleague who is proficient in scientific writing review your manuscript to eliminate grammar errors and ensure the paper is comprehensive.

Experimental design

As introduced in Section Results (lines 316, 329, 336) , there are multiple parameters used in the presented method that are decided “after several experimentations”. What is the dataset used to tune such parameters? Are these parameters decided using the two self-created datasets that are also used to evaluate the proposed method?

Validity of the findings

The authors use two self-created databases for evaluating the proposed method and for comparing it with other previously presented methods.
Having more details of the datasets can help the readers to better understand the performance of the presented method. For instance, why were these two books chosen? How many figure-captions paper in each dataset and what is the annotation process.
There are other readily available datasets for figure-caption extraction in medical documents, such as the dataset used in citation 1. How is the performance of the presented method in such databases?

---

## Round 0.2 · accepted · Accept

I'm happy with accepting the revised paper.

Reviewer 2 ·

Basic reporting

The authors addressed all the comments. Overall, the paper is well-written and clear for readers to understand.

Experimental design

No comment

Validity of the findings

No comment